# The Detection of Face-like Stimuli at the Edge of the Infant Visual Field

**DOI:** 10.3390/brainsci12040493

**Published:** 2022-04-13

**Authors:** Chiara Capparini, Michelle P. S. To, Vincent M. Reid

**Affiliations:** 1Department of Psychology, Lancaster University, Lancaster LA1 4YF, UK; m.to@lancaster.ac.uk; 2School of Psychology, University of Waikato, Hamilton 3240, New Zealand; vincent.reid@waikato.ac.nz

**Keywords:** face processing, infant perception, peripheral vision, visual field, social cognition

## Abstract

Human infants are highly sensitive to social information in their visual world. In laboratory settings, researchers have mainly studied the development of social information processing using faces presented on standard computer displays, in paradigms exploring face-to-face, direct eye contact social interactions. This is a simplification of a richer visual environment in which social information derives from the wider visual field and detection involves navigating the world with eyes, head and body movements. The present study measured 9-month-old infants’ sensitivities to face-like configurations across mid-peripheral visual areas using a detection task. Upright and inverted face-like stimuli appeared at one of three eccentricities (50°, 55° or 60°) in the left and right hemifields. Detection rates at different eccentricities were measured from video recordings. Results indicated that infant performance was heterogeneous and dropped beyond 55°, with a marginal advantage for targets appearing in the left hemifield. Infants’ orienting behaviour was not influenced by the orientation of the target stimulus. These findings are key to understanding how face stimuli are perceived outside foveal regions and are informative for the design of infant paradigms involving stimulus presentation across a wider field of view, in more naturalistic visual environments.

## 1. Introduction

The visual system of the human infant is biased to attend to stimuli that retain typical characteristics of faces. Accordingly, a preference for schematised face-like configurations over inverted ones is widely documented in newborns (e.g., [1,2]) and has been reported even before birth [3]. This predisposition to process and respond to social information can facilitate learning in social contexts, and it is a core building block of typical socio-emotional and cognitive development. This early attentional bias towards face-like configurations becomes more specific in favour of human faces with increasing age, with three-month-olds preferring photographic faces over schematised configurations [4,5]. By four months of age, face processing changes from featural to holistic, a developmental change that is initially orientation-independent and becomes specific to upright faces from seven months of age [6,7]. Similarly, preference and recognition for facial features progressively improve through perceptual learning and exposure to faces [8,9]. Over the past decades, there has been much debate over the origins of face processing, ranging from different developmental theories to models of the attentional bias for faces and face-like patterns (e.g., [10,11,12,13,14]).

Despite a significant amount of work exploring the development of social information processing, to date, the bulk of research in this field has relied on images presented between central and near-peripheral locations (up to 30° eccentricity), within the limited visual areas that can be investigated using standard computer displays. These parameters incorporate an inherent simplification of the much wider and richer natural environment in which social information spans the entire visual field. Although central visual areas have higher spatial resolution compared to peripheral vision [15], the developing human infant can nonetheless process a multitude of information presented at higher eccentricities. When looking at naturalistic scenes, humans are strongly biased towards scenes containing a person compared to those without any person present [16]. This bias is present in infants and increases from the ages of 3 to 12 months [17]. In everyday life, orienting towards a face requires more than eye movements. Often, a combination of head and eye movements is needed to detect social information and to navigate the wider surrounding visual environment. Nevertheless, studies of visual information processing have primarily been constrained to standard screens and paradigms where the head remains fixed and engaged with foveal processed space. In reality, faces not only appear in the visual field centrally but can also be presented at different spatial locations, often moving in a dynamic visual world. In this context, foveation alone is often not enough to attend to visual information and motor activity increasingly plays a role in orienting behaviours beyond near-peripheral locations [18,19]. As such, a successful visual orienting behaviour in naturalistic situations requires the infant to be active and to engage with visual information across a wide visual array. Notably, newborns move their head and eyes to maintain a face in view, more so for intact faces compared to scrambled ones [20,21]. In an interesting adaptation of these studies, one- to five-month-old infants were tested using a rotating chair that moved away from face stimuli appearing on a screen [21]. In this procedure, infants had to turn their heads to keep the visual targets in view. This paradigm demonstrated that the preferential tracking of actual face stimuli over face-like or scrambled ones declined after one month of age [21]. The authors suggested that newborn infants display orienting mechanisms favouring peripheral visual information via a predominance of subcortical structures in the brain, and this behaviour declines between one and two months of age with the maturation of cortical structures [21,22]. Despite such findings, there has been surprisingly little research on face processing at high eccentricities beyond the first postnatal months. Although reflexive orienting to peripheral targets is lessened with age, the motor abilities of the human infant become progressively refined. The ability to sit and stand allows the infants to spontaneously and freely navigate a wider visual space and to actively process a growing amount of information across their visual environment. As such, exploratory and orienting behaviours can be investigated with fewer motor constraints in older infants.

In the present study, we investigated infants’ sensitivities to face-like stimuli across the mid-peripheral visual field. Past literature suggested an expansion of the visual field extent over the first postnatal year (e.g., [23,24,25,26]). Research in the development of infant peripheral vision has often relied on the presentation of flashing lights. Whilst such stimuli can offer insight into what information infants can process across their visual fields, they are not representative of naturalistic and socially relevant stimuli, such as faces. In this study, we investigated how nine-month-old infants detect and respond to face-like stimuli displayed in their peripheral field. The infant participants were presented with face-like targets across a wide field of view covering 120° (at three eccentricities per hemifield, namely 50°, 55° and 60° to the left and right of the midline) and their orienting behaviour was measured from video recordings in a detection task. At these high eccentricities, eye movements in isolation were not enough to detect a visual target. Motor activity, as a combination of head and eye movements, was required to engage with this task. The face-like stimuli from this study were produced by spatially filtering natural images of intact faces, whilst controlling for luminance, contrast and colour to match with the simple low-level Gabor stimuli used in our prior work [27], which identified the visual limits of 9-month-olds to detect non-social stimuli to be around 50° from the line of sight. In the present study, both upright and inverted face-like stimuli were presented. Unlike Gabor patches, faces have a typical orientation and upright faces are more often encountered in our visual world. To the best of our knowledge, face orientation has never been investigated at the edge of the developing visual field and it is yet unexplored if the developmental changes in holistic face processing happen across the visual field. Hence, this study may also provide insight into the face properties that influence the ability to orient attention across a wide peripheral visual field. Further, unlike processing non-social stimuli, the processing of social stimuli had revealed a left visual field bias associated with right hemispheric dominance for faces [28,29]. This bias seems to develop between seven and eleven months of age [28]. Whether this tendency to attend more to faces in the left visual field is present even at the edge of the developing visual field and can be revealed with face-like targets is yet to be determined. If this side bias is specific to central stimulus presentation and/or to more salient face stimuli, then no side difference should be expected. Alternatively, a left visual field advantage may emerge over the right visual field. Overall, we aimed to understand whether face-like patterns can elicit different orienting behaviours and capture the infants’ attention better when compared to other non-social stimuli.

There is mixed evidence of an advantage for faces at high eccentricities in adult participants. Research with adults has found that the peripheral recognition of faces seems mostly eccentricity-dependent, with a decrease in recognition performances at increased eccentricities that is not predicted by visual acuity and size scaling according to cortical magnification, i.e., the reduced density of photoreceptors processing a visual stimulus at a more peripheral location [30,31]. Further, some neuroimaging evidence supports a face bias in favour of the central visual field, with faces activating visual areas corresponding with the central visual field as opposed to scenes that showed stronger activation in areas representing the peripheral visual field (e.g., [32,33]). Nevertheless, some authors argued that this brain differential activation is not evident if faces and buildings of a scene are rescaled according to cortical magnification [34]. Some behavioural evidence has suggested that face or building superiority at peripheral locations depends on the task demands [35]. Even though there may be a central visual field bias for faces, an advantage for peripheral face vs. other object detection has been reported at higher eccentricities [36]. Similarly, the speed of processing in a categorisation task was better for human faces presented across a wide visual field up to 80° eccentricity compared with animals and vehicles, even in crowded conditions [37]. As it stands, an advantage for faces in peripheral vision is still debatable and may depend on the task.

If there is an advantage for detecting or processing faces at high eccentricities, as some adult studies have suggested, we would expect infants to be successful at detecting peripheral schematic faces at 50° and beyond. Understanding visual field sensitivities in response to face-like configurations would enable us to determine what sort of visual information can capture infants’ attention across the visual field and, in turn, be further processed for ongoing learning.

## 2. Materials and Methods

### 2.1. Participants

Twenty-two 9-month-old infants were recruited from a large database of volunteer families living in the surroundings of Lancaster, United Kingdom. Four infants were excluded from the final sample due to fussiness (*n* = 2) and parental interference during the procedure (*n* = 2). Eighteen infants (10 females) with a mean age of 270.83 days (SD = 9.72 days; range = 8 months and 15 days to 9 months and 15 days) constituted the final sample. All infants were born at term (>37 weeks) with normal birth weight (>2.5 kg) and had no history of neurological or other medical conditions. Eye infections or vision impairments were among the criteria that excluded participation in this study. Parental informed consent was obtained for each infant prior to the beginning of the study. Participating families received GBP 10 as travel compensation and a storybook to thank them for taking part in this research. The protocol of the study was reviewed and approved by the Faculty of Science and Technology Research Ethics Committee of Lancaster University (Ethics approval reference no. FST19010). This research followed the principles and ethical standards expressed in the Declaration of Helsinki.

### 2.2. Stimuli

Stimuli consisted of either central or peripheral visual images displayed on a uniform grey background. The central stimulus was a white Gaussian blob presented within a 900 ms Gaussian temporal envelope such that 100% contrast was attained after 450 ms. Peripheral stimuli were monochromatic face-like stimuli. These stimuli were obtained from a black and white female face used in [38]. Both the upright and inverted face stimuli were adapted for this study. The original background of the face stimuli was removed and stimuli were filtered with a spatial frequency of 0.55 cycles per degree (cpd) using MATLAB (version 2018a; MathWorks, Natick, MA, USA). This specific spatial frequency was chosen as it has been already adopted to present peripheral targets at the edge of the visual field both with adults [39] and infants [27]. Peripheral targets were presented within a 900 ms Gaussian temporal envelope such that the maximal contrast was attained in the middle of their presentation time at 450 ms. Figure 1 shows the filtered face-like stimuli at maximal contrast. All stimuli subtended a 5.88° visual angle when viewed from a 40 cm distance and occupied a 180 × 180-pixel area. The stimuli were presented along the horizontal meridian. The overall luminance of the stimuli matched with the background luminance of the screen (within 25 cd/m^2^). This ensured that participants did not respond to the peripheral targets due to an abrupt luminance change. Overall, the luminance, spatial frequency, colour and contrast of the face-like stimuli matched with the Gabor stimuli presented in a prior study investigating the limits of peripheral information processing [27]. The visual presentation of each stimulus was randomly paired with an auditory tone. A total of eight beep sounds were adopted (the same sounds were used in [27]).

### 2.3. Apparatus and Procedure

Participants sat on their caregiver’s lap at approximately 40 cm from the centre of a 49-inch Samsung LC49HG90DMM curved monitor with a screen resolution of 3840 × 1080 pixels (120.30 cm width and 52.55 cm height without stand). Monitor contrast and brightness were set to 50%. This curved screen covered 126° Field Of View (FOV) in total and was placed on a table whose height was adjusted so that the infant’s line of sight corresponded to the horizontal meridian of the screen. Lights were switched off and the room was only lit by the computer screen. The infant participant’s behaviour was recorded with a hidden video camera above the screen and the experimental procedure was simultaneously recorded with a video camera behind the participant. Black felt covered the area around the monitor and the cameras to minimise light scatter. The two video cameras fed into a TV monitor and a digital video recorder that the experimenter could monitor and that were located behind a room divider. The same room set up, equipment and procedures had been used in [27]. Caregivers were instructed to keep their infant in a stable upright position and to maintain the set distance from the screen. They were also instructed to avoid any verbal and non-verbal interference with the testing procedure. The experimental procedure started with the presentation of the central stimulus for 900 ms repeated three times in quick succession, lasting 2.7 s in total. Once this central attention grabber faded away, a peripheral face-like target appeared at one of three eccentricities in the mid-peripheral visual field: 50°, 55°, 60°, either on the left or on the right visual hemifield. Each eccentricity corresponds to the visual angle between the centre of the central stimulus and the centre of the peripheral target. The peripheral target appeared three times in quick succession as the central stimulus (fading in and out 3× at 900 ms each). A total of 24 trials were presented with an inter-trial interval of 500 ms, as in [27]. Eccentricity (50°, 55°, 60°), side (left and right) and target orientation (upright and inverted) were semi-randomised across trials. The experimental procedure lasted around 4 min. The study was programmed in MATLAB using the Psychophysics Toolbox extension, version 3 [40,41].

### 2.4. Data Processing and Coding

Video recordings were coded offline to measure the orienting behaviour of the infant participant in response to the appearance of a visual target in their mid-peripheral visual field. Video recordings included the multi-camera view of the stimulus presentation and the participant’s face side by side. Videos were processed using ELAN software, version 5.9 [42,43]. For each video, a timeline was created so that each central and peripheral target presentation time (48 intervals per participant) was saved as an annotation that could be easily retrieved and evaluated without searching the stimulus from the video. At this point, each video was zoomed in to allow the coders to see only the infant’s face without being aware of the peripheral target location on each trial. First of all, trial validity was assessed by evaluating the head and eye positions for each central stimulus annotation. For a trial to be valid, the participant must be oriented towards the centre of the screen at the offset of the central stimulus presentation, before the peripheral target appears. If this condition was met, the trial was valid and the coder could proceed to the coding of the behaviour in response to the appearance of the peripheral target. The coder assessed whether the participant (a) oriented towards their left hemifield with eye and/or head movements before the next trial began; (b) oriented towards their right hemifield with eye and/or head movements before the next trial began; or (c) had no response or kept staring at the centre for the entire peripheral target presentation, looked up or down or clearly away from the screen. Instances in which the participant could orient towards both hemifields during the peripheral target presentation were rare but, if this was the case, the first clear orientation towards one hemifield was coded. We used the first behavioural response that could be coded from the neutral head and eyes position to ensure that the eccentricity value of the peripheral target was reliable. Once all the valid trials were blindly coded, the behaviours were compared with the target locations and classified either as Detection—if the direction of the orienting behaviour matched the hemifield of appearance of the peripheral target—or as No Detection—if the direction of the orienting behaviour did not match the hemifield of appearance of the peripheral target or if there was no orienting response (behaviour (c)). In this paradigm, detection indicated that the target was perceived and caused an overt behaviour. An equal number of semi-randomised trials at each location and in each hemifield controlled for random orienting behaviours and possible biases in spontaneous eye movements. The coding was accurate to the video frame level, which was 25 frames per second. While the primary blind coder judged all the video recordings, a second independent and blind coder coded a random proportion of the video recordings (8 videos, 44.44%). Cohen’s kappa was performed to determine interobserver agreement. Near perfect agreement was obtained between the two coders’ judgements, both when they evaluated trial validity and peripheral target detection. Specifically, 93% observed agreement and *k* = 0.86 were obtained judging trial validity, whereas 96% observed agreement and *k* = 0.92 were obtained judging the dependent variable of this study, namely the detection behaviour of the infant participant in response to the peripheral target presentation.

### 2.5. Analysis and Statistics

Detection rates per eccentricity and stimulus orientation were compared across participants. Data were analysed using generalised linear mixed models (GLMMs) with the lme4 package [44] implemented in R, version 3.5.2 [45]. GLMMs have provision to consider the multilevel structure of the data set and, importantly, to incorporate the unbalanced structure of the repeats following the exclusion of invalid trials (for advantages in using GLMMs, see [46]). Detection rates were considered as a dependent measure, with log odds of detection as the model outcome. To account for the within-subject design, participant number was included as a random factor. In addition, Sex was included in the model as a categorical fixed factor at the individual level, whereas Eccentricity, Side and Stimulus orientation were included as categorical fixed factors at the trial level. Confidence intervals (95% CI) were calculated using the Wald method. Models were evaluated from the simplest to the more complex and were compared by the Likelihood Ratio Test (LRT) and Akaike Information Criterion (AIC).

## 3. Results

A total of 206 valid observations were obtained, with an average of 11.44 (47.67%) valid trials per participant. Notably, 100% of the investigated orientation behaviours included a combination of eye and head movements that allowed the infant to orient towards the peripheral targets appearing at very high eccentricities. Typically, the saccadic eye movement began slightly before the start of the head movement (55.48% of orientation behaviours) or the eye and head movements were concurrent with each other (43.84% of orientation behaviours). A propensity to move the head before the eyes was extremely rare (0.68% of orientation behaviours). Individual differences played a role in movement timings, with large between-subject variability of eye and head movement tendencies, although this had no relationship with detection outcomes. In fact, the proportion of orientation behaviours in which the eye component preceded the head component ranged from a maximum of 100% to a minimum of 9% of individual orienting behaviours.

We evaluated what factors improved detection rates of peripheral face-like stimuli. The null model only including random effects (AIC = 283.15, LogLik = −139.57) was significantly improved by the inclusion of Eccentricity as a fixed effect (AIC = 258.25, LogLik = −125.13, *X*^2^(2) = 28.89, *p* < 0.001). The inclusion of both Eccentricity and Side as fixed effects further improved the Eccentricity model (AIC = 255.96, LogLik = −122.98, *X*^2^(1) = 4.29, *p* = 0.04). Including Eccentricity, Side and Orientation did not improve the Eccentricity and Side model (AIC = 257.89, LogLik = −122.94, *X*^2^(1) = 0.07, *p* = 0.79). Similarly, including Eccentricity, Side, Orientation and Sex did not improve the Eccentricity and Side model (AIC = 258.01, LogLik = −122.01, *X*^2^(1) = 0.07, *p* = 0.79). An Eccentricity per Side interaction term was also added to the model but it did not improve the Eccentricity and Side model (AIC = 257.80, LogLik = −121.90, *X*^2^(2) = 2.16, *p* = 0.34). Thus, we retained the model with lowest AIC that provided the most parsimonious explanation of the data, namely the Eccentricity and Side model (Table 1).

Results revealed that detection rates were mostly explained by target eccentricity, with detection rates dropping at high eccentricities, particularly at 60°. Targets appearing at 50° and 55° both led to good detection performances that did not significantly differ from each other, with targets appearing at 55° (*M* = 66.10, SE = 6.47) expected to have a 0.59 unit decrease in log odds of detection compared to 50° (*M* = 77.80, SE = 5.40, 95% CI = −1.35, 0.18). Performances significantly dropped at 60° (*M* = 31.80, SE = 6.38), with a 2.01 unit decrease in log odds of detection compared to targets appearing at 50° (95% CI = −2.81, −1.22). In addition, Side had a partial role in explaining the detection rates results. Overall, better detection rates were obtained for targets appearing on the left side. In fact, targets appearing on the right side (*M* = 51.40, SE = 6.00) were expected to have a 0.66 unit decrease in log odds of detection compared to targets appearing on the left side (*M* = 67.10, SE = 5.75, 95% CI = −1.29, −0.03; Figure 2).

The contribution of valid trials was similar across eccentricities and sides and this factor did not explain these results. In more detail, 71 valid trials were obtained at 50° (31 on the left and 40 on the right side), 68 valid trials were obtained at 55° (34 left and 34 right trials) and 67 valid trials were obtained at 60° (33 left and 34 right trials). A few more valid trials on the right side were localised at 50° where detection performance was at its peak. Thus, this could have favoured detection rates on the right side and does not explain better overall performances on the left side. In this model, the proportion of the outcome variance due to individual differences, as expressed by the Interclass Correlation Coefficient (ICC), was 5.77%. Post hoc analyses were conducted on the effects that significantly influenced the model. Post hoc pairwise comparisons with Tukey’s adjustment for multiple comparisons confirmed no difference between detection rates of peripheral targets appearing at 50° and 55° (95% CI = −0.33, 1.50), whereas significant differences emerged between targets appearing at 50° and 60° (95% CI = 1.06, 2.97), and also between 55° and 60° (95% CI = 0.52, 2.34). Further, detection rates of targets appearing on the left side marginally differed from targets on the right side (95% CI = 0.03, 1.29).

## 4. Discussion

This work aimed to explore face-like stimuli detection at the edge of the developing visual field. The developmental mechanisms behind face detection and processing have thus far been mostly investigated between central and near-peripheral locations, with exploration at high eccentricities absent from the literature. We presented 9-month-old infants with schematic face patterns appearing in the mid-peripheral visual field (50°, 55° and 60°, to the left and right side of a central location) in a detection and orientation task. Our results suggested that detection is heterogeneous across the visual field, with detection rates mostly explained by eccentricity. Specifically, detection progressively decreased with increasing eccentricities, with good detection rates from 50° to 55° and a significant drop at 60°. Overall, detection performances were slightly better for targets appearing in the left hemifield.

In everyday life, we deal with faces entering the visual field at various locations. A combination of head and eye movements is often required to bring faces into foveal view, enabling further processing and social interaction. Nevertheless, most laboratory paradigms have studied the perceptual and cognitive mechanisms behind this social orienting behaviour using standard computer displays where eye movements alone could process the limited visual space under investigation. In the present task, infants always combined head and eye movements to orient towards the peripheral targets appearing at high eccentricities. In line with past adult data [18,19], the timing of the eye and head components of the orienting behaviour showed large variability between subjects. Further, the current study showed that 9-month-old infants could detect face-like configurations at high eccentricities, up to about 55° in their visual field. The decreasing detection rates at larger eccentricities may be explained by a developing peripheral vision. Visual acuity in the mid-peripheral visual field does not seem fully mature, as the developed adult visual field extends to around 95–110° eccentricity [39]. This is in line with past evidence that reported peripheral vision developed across the course of the first postnatal year (e.g., [23,25]). It is of note that the current stimuli were carefully matched with the background luminance and the stimuli faded in and out, gradually reaching maximal contrast. As such, in this paradigm the orientation could not be due to abrupt low-level changes in the visual field such as luminance changes, flashing or flickering of the stimuli. It is possible that more salient images—such as coloured photographs of real faces—can be detected even further in the visual field, but utilising those in the present study would have confounded low- and high-level visual feature processing with the consequence that we would not be able to make conclusions concerning the drivers behind infant orienting behaviours.

Prior perimetry studies have reported mixed results in terms of the extent and characteristics of the developing visual field. Some studies have implied a developing peripheral vision during the entire first postnatal year [23], whereas other studies have reported adult-like performances around six months of age [26,47]. This is likely due to different testing procedures and stimuli adopted across studies, with most studies using highly salient stimuli such as flashing lights and variable procedures. In the present study, we paid particular attention to the low-level visual features of the peripheral targets to ensure that detection was not guided by low-level abrupt changes in the visual field. Further, this study adopted the same testing procedures, paradigm and low-level visual features of the peripheral targets (colour, contrast, spatial frequency and luminance) already used in a prior study presenting Gabor patches as peripheral targets [27]. Even though both investigations found heterogeneous performances across the developing visual field and decreased detection performances at higher eccentricities, detection was better with schematic faces than with Gabor patches at each investigated mid-peripheral eccentricity. In particular, while detection rates dropped below chance with Gabor patches presented at 55° [27], the drop was at 60° using face-like targets. This would suggest that, although peripheral vision is still developing, stimulus type can enhance detection and attentional mechanisms even at the edge of the developing visual field. Given that the low-level visual features of the peripheral targets were comparable, the structural configuration of the face patterns likely played a relevant role in detection performance.

This face advantage beyond foveal presentations is in line with past adult behavioural evidence, such as a more efficient detection of peripheral faces compared with objects or other animals [37]. Intriguingly, the authors investigated high eccentricities up to 80°, whereas most studies investigating peripheral processing did not go beyond near-peripheral locations (up to 30° eccentricity). The current data provide evidence of a pop-out attentional response to faces at high eccentricities even during infancy. This advantage may be due to a rapid pre-attentive mechanism for faces that is activated prior to orientation and fixation, as suggested by past findings [16,36]. At the neural level, this is likely the same subcortical visual pathway that enables rapid detection and reaction to faces and threatening social information, which is evident in the newborn [22,48] and has residual activity in adults [48,49,50]. Such a mechanism is critical to quickly detect biologically relevant information and guide the infant’s exploratory behaviour.

Interestingly, infants’ detection performances were not influenced by the orientation of the target stimulus. At high eccentricities, both upright and inverted face-like stimuli were equally salient and yielded comparable detection rates, independent of their orientation. This finding is in line with past infant research which theorised that initial orienting mechanisms and attention-getting mechanisms are not influenced by face orientation [51]. The authors presented 6-month-old infants with circular visual arrays including faces and other objects and found a pop-out effect for faces compared to non-face objects. The first look direction towards faces was not influenced by face orientation, with both upright and inverted faces attracting participants’ first looks. In contrast to this, the authors found an orientation effect when measuring attention-holding mechanisms, i.e., those mechanisms involved in maintaining the infant’s attention once the visual target has been detected [51]. As such, an orientation effect could be evident from measures such as looking duration, number of fixations or saccadic reaction times (see, for instance, [51] for an investigation using the number of fixations in infancy, [52] for a study measuring saccadic latencies in infancy or [49] for a study using saccadic reaction times with adults). In the present study, looking duration and other measures in the time domain were not easily assessable as both clear fixations towards the target on screen and more general orientations towards the correct hemifield indicated detection. Future studies may address whether attention-holding mechanisms show an advantage for upright face-like patterns compared to inverted patterns even at high eccentricities. As it stands, we provide evidence that orienting to faces appearing at the edge of the visual field is guided by some general facial structural features; this is consistent with past infant findings on a face orientation bias at near-peripheral locations [51]. Peripheral vision has a role in rapid orienting towards changes and threats entering the visual field. As such, a rapid orienting response that is not too selective in terms of stimulus orientation may be an optimal solution before faces are further processed. Research with adults has also found no difference in both accuracy and in target-directed saccades to upright and inverted faces presented at peripheral locations in circular arrays [53]. An accuracy difference in favour of upright faces emerged only following some practice with upright faces before the task [53]. Similarly, past research found a happy-face advantage in adults’ peripheral vision that was not affected by stimulus inversion [54].

The current investigation also showed that detection performances are slightly influenced by the target side, with increased detection rates for filtered faces appearing in the left hemifield compared to the right hemifield. This does not seem to be explained by the contribution of valid trials by side and eccentricity. Interestingly, this side effect was not evident when presenting infants with Gabor patches in the same paradigm [27]. This behavioural advantage for face-like stimuli appearing in the left visual field is in line with a left visual field bias for face processing developing within the first postnatal year [28,55] and may reflect an emerging functional cerebral dominance in the right hemisphere (e.g., [29,56,57,58]). Past research has mostly focused on an early bias to attend towards the left hemiface [28,55], which has also been linked with left social positioning biases and cradling behaviours that favour right hemisphere processing [59,60,61]. Interestingly, the present study provided some preliminary indication of lateralisation of face processing at high eccentricities, beyond centrally presented faces and side biases within the face. Although side had a marginal role compared to eccentricity in explaining the detection rates, this bias may emerge from or soon after 9 months of age [28,62]. Future investigations may address whether this effect is more pronounced in older infants and whether there may be a neural correlate of a left visual field superiority for detection of face patterns at the edge of the developing visual field.

## 5. Conclusions

Sensitivities to face-like stimuli are unequal at the edge of the visual field of 9-month-old infants. Detection progressively decreased at higher eccentricities, with successful detection rates up to about 55° in the mid-peripheral visual field. Infant performance further suggested a marginal influence of target side, with an advantage for stimuli appearing in the left hemifield. Compared with a past investigation adopting identical procedures and low-level visual features of the non-face peripheral targets, the present findings revealed increased peripheral sensitivities to face-like stimuli. Overall, this work suggests that the spatial boundaries of infant visual attention may be influenced by the stimulus type even at very high eccentricities. These findings may be informative in developing future studies investigating wide natural scenes. In conclusion, this is the first study to identify the limits of infants’ peripheral information processing utilising face-like stimuli across a wide visual field of over 120°. The results show that peripheral sensitivities to face-like targets are mostly predicted by eccentricity.

## Figures and Tables

**Figure 1 brainsci-12-00493-f001:**
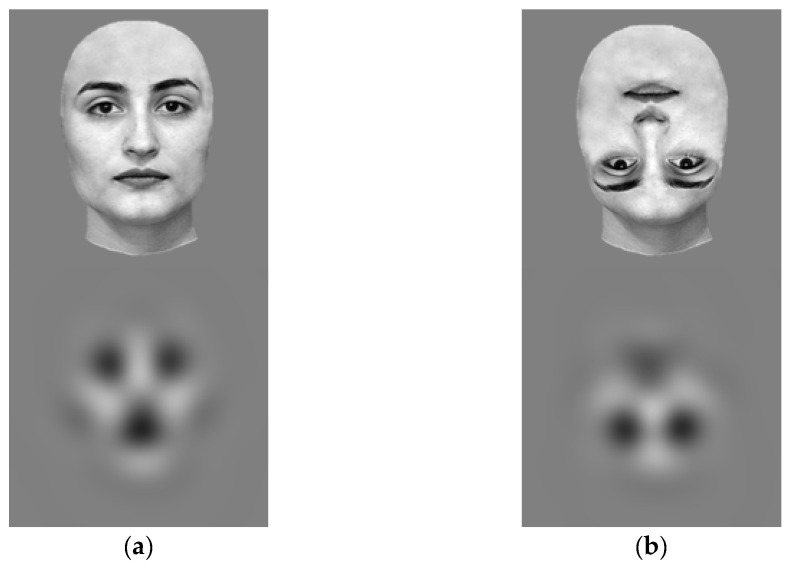
The upright (**a**) and inverted (**b**) face-like stimuli presented as peripheral targets (**bottom row**). Stimuli were obtained by filtering the upright and inverted faces used in [38] (**top row**) with a spatial frequency of 0.55 cpd.

**Figure 2 brainsci-12-00493-f002:**
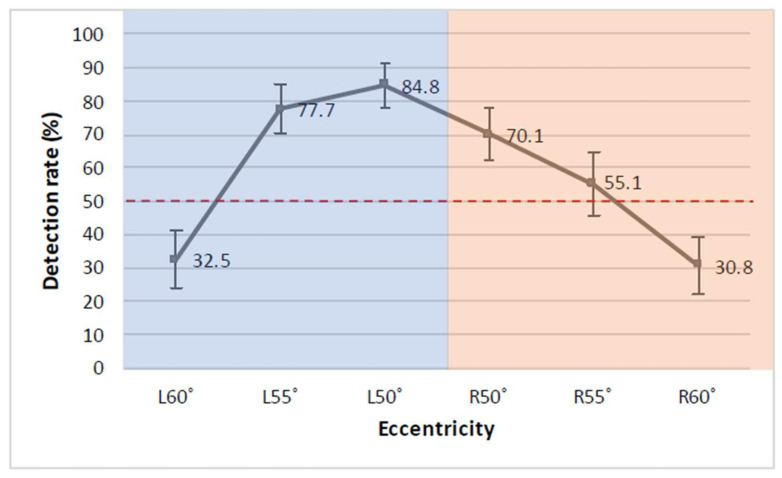
The detection rates of face-like targets across eccentricities in the left (blue) and right (orange) hemifields. Error bars represent +/− 1 SE.

**Table 1 brainsci-12-00493-t001:** The generalised linear mixed effects model (GLMM) results.

Fixed Effects					
	Estimate	SE	95% CI	Z	*p*
Intercept	1.58	0.37	0.85, 2.31	4.24	<0.001 ***
Eccentricity (55°)	−0.59	0.39	−1.35, 0.18	−1.50	0.13
Eccentricity (60°)	−2.01	0.41	−2.81, −1.22	−4.95	<0.001 ***
Side (Right)	−0.66	0.32	−1.29, −0.03	−2.04	0.04 *
**Random Effects**					
	**Variance**	**SD**		**95% CI**	
Intercept	0.20	0.45		0, 1.04	

Note. Significance code *** *p*-value [0, 0.001], * *p*-value [0.01, 0.05]. Confidence intervals calculated using the Wald method. R model equation: Detection ~ Eccentricity + Side + (1|Participant).

## Data Availability

The data presented in this study are available on request from the corresponding author.

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
