# Peer review of "The Detection of Face-like Stimuli at the Edge of the Infant Visual Field"

_brainsci, 2022, doi:10.3390/brainsci12040493_

Round 1

Reviewer 1 Report
In my opinion, this work is very interesting and I agree that it fills a gap in the literature investigating the development of face perception in infants. In fact, authors found the lateral boundaries of face detection (traceable between 55° and 60°, at which detection rates significantly drop below the chance level) in 9mo infants, leading the way to further studies aimed to reveal the developmental trajectory of this ability in humans. Moreover, they showed a sort of early LVF (left visual field) advantage for face detection, especially for lower and medium eccentricities.
The manuscript is very well-written, the objectives are interesting and reasonable, tha sample size is adequate, methods and procedures are complete (despite the evident setting difficulties) and well described, the analyses are appropriate and sound. So I have little to say and I will raise some minor issues and curiosities:
Introduction
- As authors used inverted faces in their experiment, the first part of the introduction might get benefit if authors add a short passage on the development of the proficiency in the configural (vs. featural) processing of upright faces and on the timing of its emergence. For example, some authors found that such a skill starts from the 4/5th mo, gradually developing over the 1st y [e.g., Cashon, C. H., & Holt, N. A. (2015). Chapter Four—Developmental Origins of the Face Inversion Effect. In J. B. Benson (Ed.), Advances in Child Development and Behavior (Vol. 48, pp. 117–150). JAI. https://doi.org/10.1016/bs.acdb.2014.11.008].
- Moreover, I suggest that authors mention in the introduction the topic of the side-biases for face/social stimuli in infanthood [e.g., this is a very recent study in which eye-tracking has been used in order to show early lateral biases in social gaze: Davis, R., Donati, G., Finnegan, K., Boardman, J. P., Dean, B., Fletcher‐Watson, S., & Forrester, G. S. (2022). Social gaze in preterm infants may act as an early indicator of atypical lateralization. Child Development. https://doi.org/10.1111/cdev.13734]
Materials and methods
- Can the authors better explain why they used only 24 trials (meaning only 2 occurrences for each condition)?
Results
- Maybe I missed something, but I wondered why authors did not report post-hoc comparisons between detection rates at different left vs. right eccentricities (for example, I would be interested to know significant differences between the datapoints authors showed in Figure 2).
Discussion
- While other results are well discussed, the left-hemifield advantage is here only mentioned by the authors. However, I think this is a very interesting finding. In fact, very few studies on functional lateralization have explored this particular peripheral paradigm, most of them using a divided visual field task, in which the stimulus is quite close to the center of the screen (therefore, with smaller eccentricity degrees). Instead, the larger eccenticities used in this study can be relevant cues of individuals' hemispheric dominance. I think authors should discuss more deeply this result to the benefit of the scholars interested in the development of human lateralization. In particular, it should be noted that, during the first months of their life, infants' whole social environment is their mother, who cradle them in their arms for most of the time. Interestingly, most of mothers hold their infants on the left side of their body and with the left arm, regardless of their handedness .
Starting from this positioning, we could assume that, in the first months, infants have constantly their mother's face in their left (maube peripheral) hemifield, which constitutes the largest part of their socio-emotional environment. In this regard, it has very recently been proposed that this left-cradling bias in itself might canalize the infants' typical neurodevelopment through the lateralized exposure of their sensorymotor system to socio-emotional stimuli in the eraly stage of brain development 
In fact, it has been shown that cradling infants on the left rather than on the right can provide infants with emotional information of better quality and is associated with the later development of social competencies in adulthood that are typically lateralized (such as the population-level LVF for face perception;  I think that including this studies might enrich authors' discussion of the presence of such an asymmetry in their results.
- Finally, as authors interestingly used a face inversion paradigm (although no significant result was found), they should refer very quickly to the possible reasons why no difference was shown between upright and inverted faces. In fact, no difference, in the face inversion paradigm, is a result per se. Is it possible to conclude that infants have not yet fully developed a configural processing of faces? Otherwise, might this lacking of the face inversion effect be due to the spatial frequency filters used for the face stimuli?

Author Response
Dear Reviewer 1,
Many thanks for the comments on our manuscript ID brainsci-1636771 entitled "The
detection of face-like stimuli at the edge of the infant visual field" submitted to Brain
Sciences.
A point-by-point response (in bold) to each comment can be found below. Text added in the
manuscript is reported in blue font colour. The lines and pages in our response refer to the
new manuscript version with tracked changes.
We hope that these changes will be to your satisfaction. We express our sincere gratitude
for your comments and your time.
Sincerely,
Chiara Capparini and colleagues

In my opinion, this work is very interesting and I agree that it fills a gap in the literature
investigating the development of face perception in infants. In fact, authors found the
lateral boundaries of face detection (traceable between 55° and 60°, at which detection
rates significantly drop below the chance level) in 9mo infants, leading the way to further
studies aimed to reveal the developmental trajectory of this ability in humans. Moreover,
they showed a sort of early LVF (left visual field) advantage for face detection, especially for
lower and medium eccentricities.
The manuscript is very well-written, the objectives are interesting and reasonable, the
sample size is adequate, methods and procedures are complete (despite the evident setting
difficulties) and well described, the analyses are appropriate and sound. So I have little to
say and I will raise some minor issues and curiosities:
We thank the Reviewer for recognising that this is an interesting work and fills a
gap in the literature about infant face perception. Also, thanks for stating that
objectives, methods and results are complete and appropriate.
Introduction
- As authors used inverted faces in their experiment, the first part of the introduction might
get benefit if authors add a short passage on the development of the proficiency in the
configural (vs. featural) processing of upright faces and on the timing of its emergence. For
example, some authors found that such a skill starts from the 4/5th mo, gradually
developing over the 1st y [e.g., Cashon, C. H., & Holt, N. A. (2015). Chapter Four—
Developmental Origins of the Face Inversion Effect. In J. B. Benson (Ed.), Advances in Child
Development and Behavior (Vol. 48, pp. 117–150). JAI.
https://doi.org/10.1016/bs.acdb.2014.11.008].
- Moreover, I suggest that authors mention in the introduction the topic of the side-biases
for face/social stimuli in infanthood [e.g., this is a very recent study in which eye-tracking
has been used in order to show early lateral biases in social gaze: Davis, R., Donati, G.,
Finnegan, K., Boardman, J. P., Dean, B., Fletcher‐Watson, S., & Forrester, G. S. (2022). Social
gaze in preterm infants may act as an early indicator of atypical lateralization. Child
Development. https://doi.org/10.1111/cdev.13734]
We thank the Reviewer for providing some suggestions to expand the
introduction. We have added some information and references about the
configural and featural face processing development in the first paragraph, citing
the work of Cashon and colleagues as suggested. Please see lines 33-36:
By four months of age face processing changes from featural to holistic, a
developmental change that is initially orientation-independent and becomes
specific to upright faces from seven months of age [6, 7].
We also welcome the suggestion of introducing early side biases for face stimuli in
the introduction, as now reported at lines 116-124:
Further, unlike processing non-social stimuli, the processing of social stimuli
had revealed a left visual field bias associated with right hemispheric
dominance for faces [28, 29]. This bias seems to develop between seven and
eleven months of age [28]. Whether this tendency to attend more to faces in
the left visual field is present even at the edge of the developing visual field
and can be revealed with face-like targets is yet to be determined. If this side
bias is specific to central stimulus presentation and/or to more salient face
stimuli then no side difference should be expected. Alternatively, a left visual
field advantage could emerge over the right visual field.
Materials and methods
- Can the authors better explain why they used only 24 trials (meaning only 2 occurrences
for each condition)?
We thank the Reviewer for raising this point. We used 24 trials to adopt the same
procedure of a previous study we conducted with non-social stimuli (Gabor
patches) and with an identical set up and procedure, which is currently in press in
Developmental Psychobiology. Before implementing this initial study, we had
tested some pilot participants and found that 24 was an ideal number of trials. This
choice took into consideration both the infants’ limited attention span and the
non-salient nature of our stimuli, with the procedure in line with the psychophysics
tradition that we have used. Although having a higher amount of trials would have
been ideal, we compensated for the unbalanced structure of the data using
generalised linear mixed effects models.
Results
- Maybe I missed something, but I wondered why authors did not report post-hoc
comparisons between detection rates at different left vs. right eccentricities (for example, I
would be interested to know significant differences between the datapoints authors
showed in Figure 2).
We thank the Reviewer for raising this point. We reported the post-hoc
comparisons of the effects that significantly influenced the model, namely the
main effect of eccentricity and the main effect of side. The model that better
explained the data was the one without interactions (please see lines 302-306),
hence we did not run these post-hoc tests. We have also specified this in the
revised manuscript, lines 337-338:
Post hoc analyses were conducted on the effects that significantly influenced the
model.
Discussion
- While other results are well discussed, the left-hemifield advantage is here only mentioned
by the authors. However, I think this is a very interesting finding. In fact, very few studies on
functional lateralization have explored this particular peripheral paradigm, most of them
using a divided visual field task, in which the stimulus is quite close to the center of the
screen (therefore, with smaller eccentricity degrees). Instead, the larger eccenticities used
in this study can be relevant cues of individuals' hemispheric dominance. I think authors
should discuss more deeply this result to the benefit of the scholars interested in the
development of human lateralization. In particular, it should be noted that, during the first
months of their life, infants' whole social environment is their mother, who cradle them in
their arms for most of the time. Interestingly, most of mothers hold their infants on the left
side of their body and with the left arm, regardless of their handedness. Starting from this
positioning, we could assume that, in the first months, infants have constantly their
mother's face in their left (maybe peripheral) hemifield, which constitutes the largest part of
their socio-emotional environment. In this regard, it has very recently been proposed that
this left-cradling bias in itself might canalize the infants' typical neurodevelopment through
the lateralized exposure of their sensorimotor system to socio-emotional stimuli in the early
stage of brain development. In fact, it has been shown that cradling infants on the left
rather than on the right can provide infants with emotional information of better quality
and is associated with the later development of social competencies in adulthood that are
typically lateralized (such as the population-level LVF for face perception; I think that
including these studies might enrich authors' discussion of the presence of such an
asymmetry in their results.
We thank the Reviewer for suggesting to further discuss the left visual field bias
and for providing this interesting point about the left cradling bias. Firstly, we have
expanded the left visual field bias and our expectations in the introduction (please
see the above point). Also, we have elaborated this aspect further in the
discussion, lines 450-480:
This behavioural advantage for face-like stimuli appearing in the left visual field
is in line with a left visual field bias for face processing developing within the
first postnatal year [28, 55] and could reflect an emerging functional cerebral
dominance in the right hemisphere (e.g. [29, 56-58]). Past research has mostly
focused on an early bias to attend towards the left hemiface [28, 55], which
has also been linked with left social positioning biases and cradling behaviours
that favor right hemisphere processing [59-61]. Interestingly, the present
study provided some preliminary indication of lateralization of face processing
at high eccentricities, beyond centrally presented faces and side biases within
the face. Although side had a marginal role compared to eccentricity in
explaining the detection rates, this bias could possibly emerge from or soon
after 9 months of age [28, 62]. Future investigations may address whether this
effect is more pronounced in older infants and whether there may be a neural
correlate of a left visual field superiority for detection of face patterns at the
edge of the developing visual field.
- Finally, as authors interestingly used a face inversion paradigm (although no significant
result was found), they should refer very quickly to the possible reasons why no difference
was shown between upright and inverted faces. In fact, no difference, in the face inversion
paradigm, is a result per se. Is it possible to conclude that infants have not yet fully
developed a configural processing of faces? Otherwise, might this lacking of the face
inversion effect be due to the spatial frequency filters used for the face stimuli?
The orientation null result is an interesting point. For this reason, we have
discussed this quite extensively from lines 415 to 445. Our hypothesis is that this
may be due to the task we adopted, which investigated attention-getting
mechanisms (whether the infant detected a peripheral target or not). Past
evidence suggests that an orientation effect should be evident in attention-holding
mechanisms (for instance, using looking duration or number of fixations to the
target). It may be also worth exploring whether the specific low-level visual
features of the targets had an influence, as the Reviewer suggested. In this regard
there is limited work exploring high eccentricities. We have now provided some
possible explanations considering adult data together with some potential future
directions. 

Reviewer 2 Report

The current manuscript investigated eighteen 9-month-old infants’ visual sensitivity to face-like stimuli at various eccentricities in the visual field. The novelty of the work is clearly presented by the authors, and I think the study represents an important addition the face perception literature in infancy. I have one major concern and a few minor concerns related to the manuscript. The major concern centers around the authors conclusion and discussion of a marginally significant effect (i.e., targets on the left side were marginally differed from targets on the right side). Reporting this effect in the results section is okay; however, line 297-298 the authors make a general statement that overall processing is better on the left. This conclusion is made from non-significant results and needs to be removed or restated. This result is also the focus of the paragraph starting on line 381. This paragraph mentions a “slight influence” which is a more accurate representation. Yet, it is putting a lot of energy and space into a non-significant effect. Again, it is fine to include this pattern of observation in the results but such a strong statement in the discussion section is not warranted by the data in the manuscript. If the authors expected a target side difference and this result was in line with the expectation, then this pattern should have been established in the introduction. This was not done so it appears this was an non-hypothesized effect, further indicating the marginal result should not be discussed at length until further research can be done to investigate the pattern. On a positive note, the discussion section does provide an excellent summary of previous orientation studies in infancy to potentially explain the current experiments lack of an inversion effect. Minor: Typo Ln 55 (five-month-old infants) Table/Figure formatting needs to be updated to journals standards.

Author Response

 Dear Reviewer 2,

Many thanks for the comments on our manuscript ID brainsci-1636771 entitled "The detection of face-like stimuli at the edge of the infant visual field" submitted to Brain Sciences.

A point-by-point response (in bold) to each comment can be found below. Text added in the manuscript is reported in blue font colour. The lines and pages in our response refer to the new manuscript version with tracked changes.

We hope that these changes will be to your satisfaction. We express our sincere gratitude for your comments and your time.

Sincerely,

Chiara Capparini and colleagues

The current manuscript investigated eighteen 9-month-old infants’ visual sensitivity to face-like stimuli at various eccentricities in the visual field. The novelty of the work is clearly presented by the authors, and I think the study represents an important addition the face perception literature in infancy.
We thank the Reviewer for recognising the novelty of this work and for highlighting the addition it would make to the face perception literature.

I have one major concern and a few minor concerns related to the manuscript. The major concern centers around the authors conclusion and discussion of a marginally significant effect (i.e., targets on the left side were marginally differed from targets on the right side). Reporting this effect in the results section is okay; however, line 297-298 the authors make a general statement that overall processing is better on the left. This conclusion is made from non-significant results and needs to be removed or restated. This result is also the focus of the paragraph starting on line 381. This paragraph mentions a “slight influence” which is a more accurate representation. Yet, it is putting a lot of energy and space into a non-significant effect. Again, it is fine to include this pattern of observation in the results but such a strong statement in the discussion section is not warranted by the data in the manuscript. If the authors expected a target side difference and this result was in line with the expectation, then this pattern should have been established in the introduction. This was not done so it appears this was a non-hypothesized effect, further indicating the marginal result should not be discussed at length until further research can be done to investigate the pattern. On a positive note, the discussion section does provide an excellent summary of previous orientation studies in infancy to potentially explain the current experiments lack of an inversion effect.
We thank the Reviewer for raising this point about the marginal side effect. We generally agree with the Reviewer and for this reason we have not given too much space to the marginally significant side effect in the discussion. First of all, we took the opportunity to better specify our expectations about lateralisation in the introduction. In particular, while we had some hypotheses about eccentricity and this was our main focus, we had no past data at such high eccentricities for orientation and side and we could only have some conjectures. Both an absence of side effects at very high eccentricities and a left visual field bias as reported at more central locations were plausible. This is now reported in the introduction, lines 116-124:

Further, unlike processing non-social stimuli, the processing of social stimuli had revealed a left visual field bias associated with right hemispheric dominance for faces [28, 29]. This bias seems to develop between seven and eleven months of age [28]. Whether this tendency to attend more to faces in the left visual field is present even at the edge of the developing visual field and can be revealed with face-like targets is yet to be determined. If this side bias is specific to central stimulus presentation and/or to more salient face stimuli then no side difference should be expected. Alternatively, a left visual field advantage could emerge over the right visual field.

Even though eccentricity mostly explained the detection performance, side had also a partial role in the model. This role can be marginal compared to eccentricity but it is not a ‘non-significant effect’ as stated by the Reviewer. Given that this effect and post hoc comparison were reported in the results and reached significance, we believe that they should be also discussed to a certain degree. The specific statement that the Reviewer pointed out has now been rephrased and we now say that performances were “slightly better” for targets appearing in the left hemifield (line 356).
At the same time, another Reviewer said that the lateralisation aspect was very interesting and we should expand this. We have not elaborated on this topic too much. We now highlight some studies indicating that a left visual field bias could emerge from or soon after nine months of age (the age we investigated). For this reason, we suggested future work with older infants, lines 475-480:

Although side had a marginal role compared to eccentricity in explaining the detection rates, this bias could possibly emerge from or soon after 9 months of age [28, 62]. Future investigations may address whether this effect is more pronounced in older infants and whether there may be a neural correlate of a left visual field superiority for detection of face patterns at the edge of the developing visual field.

Minor: Typo Ln 55 (five-month-old infants) Table/Figure formatting needs to be updated to journals standards.

We thank the Reviewer for noticing a typo that has been corrected in the revised manuscript. Regarding formatting, we used the Brain Sciences template and had not changed the table/figure formatting. We are not sure with respect to what the Reviewer refers to and how we could improve the formatting. We have moved Figure 2 closer to its mention in the text.

Reviewer 3 Report

  1. Absence of eye-tracking data: It is understandable that eye-tracking and gaze data in infants will be challenging, but this will provide more quantitative evidence of the infants' gaze.
  2. The paper states, “These findings are key to understanding how face stimuli are perceived outside foveal regions and are informative for the design of infant paradigms involving stimulus presentation across a wider field of view, in more naturalistic visual environments.” However, the experiment design lacks the control stimuli (non-face stimuli). In addition, the infants' performance was not influenced by the orientation of the target stimulus (line 350). Behavioral and neurophysiological studies of primates (humans and non-human primates) demonstrate that upright faces have a higher perceptual valence than inverted faces. The paper argues that initial orienting mechanisms and attention-getting mechanisms are unaffected by face orientation based on previous research (lines 353-354). These stimulus-aspecific detection results contradict the primary claim made in the paper.
  3. There are no control (non-face) experiments in the paper, so its scope is limited to peripheral visual field testing in infants. Moreover, the paper doesn't elaborate on the mechanism of non-foveal stimuli perception (lines 20-21) except for a mention of detection performance.
  4. It is recommended that the paper limit its claims and scope proportionally to the results and experiment design.

Author Response

 Dear Reviewer 3,

Many thanks for the comments on our manuscript ID brainsci-1636771 entitled "The detection of face-like stimuli at the edge of the infant visual field" submitted to Brain Sciences.

A point-by-point response (in bold) to each comment can be found below. The lines and pages in our response refer to the new manuscript version with tracked changes.

We hope that these changes will be to your satisfaction. We express our sincere gratitude for your comments and your time.

Sincerely,

Chiara Capparini and colleagues

  1. Absence of eye-tracking data: It is understandable that eye-tracking and gaze data in infants will be challenging, but this will provide more quantitative evidence of the infants' gaze.

We thank the Reviewer for raising this point. It is something that we have evaluated and we actually implemented an eye-tracking procedure to test infants in a wide visual area (most commercial eye-tracking solutions can track screens up to 30 inches, whereas our monitor was 49 inches and covered more than 120deg). Although we have been using this for some research paradigms, this was not the case of this study for an array of reasons. Firstly, when we implemented this specific eye-tracking solution we realised that it was not suitable for paradigms in which the stimuli are not very salient. In fact, for this study we specifically aimed to present peripheral stimuli that did not cause an abrupt change in terms of low-level visual features and that were not highly salient compared to the background (please see stimuli description in subsection 2.2 Stimuli). In this context, the infrared light of the eye-tracker solution we implemented to track such a wide visual filed was attracting the infant’s attention and caused distraction between the central stimulus and the peripheral target appearance. Furthermore, the orienting behaviour in this study is so overt (considering that infants had to move both their eyes and head in order to detect targets across 120deg field of view) that eye-tracking data would make little contribution beyond the data that we could acquire related to overt behaviours. Thus, because of the extent of the investigated visual field and the overt orienting behaviour we measured, we believe that the results stand on their own.

  1. The paper states, “These findings are key to understanding how face stimuli are perceived outside foveal regions and are informative for the design of infant paradigms involving stimulus presentation across a wider field of view, in more naturalistic visual environments.” However, the experiment design lacks the control stimuli (non-face stimuli). In addition, the infants' performance was not influenced by the orientation of the target stimulus (line 350). Behavioral and neurophysiological studies of primates (humans and non-human primates) demonstrate that upright faces have a higher perceptual valence than inverted faces. The paper argues that initial orienting mechanisms and attention-getting mechanisms are unaffected by face orientation based on previous research (lines 353-354). These stimulus-aspecific detection results contradict the primary claim made in the paper.

We thank the Reviewer for raising this point and for allowing us to better explain that we do have data with non-social control stimuli. As mentioned in the manuscript, this study was previously conducted with non-social stimuli (Gabor patches) with infants of the same age, identical set up and procedures. All the low-level visual features of the stimuli (colour, spatial frequency, contrast, luminance) were the same, with the only difference being that face-like stimuli had two orientations. We have now clearly specified this in the methods (please see lines 216-217). This control study is currently in press in Developmental Psychobiology (doi will be 10.1002/dev.22274 and we would be happy to provide this via the Editor if not yet available online when the Reviewer reads our response).
The Reviewer also made an interesting point about face-like stimuli orientation. As noted, in the first paragraph of the introduction we mentioned that upright faces have a higher perceptual valence than inverted faces already from early developmental stages. The Reviewer said that our results contradict this claim. This may be true when central to near-peripheral visual areas are investigated. But does this attentional bias hold when faces are presented at the edge of the infant visual field? We presented mixed evidence of, and advantage of, upright faces at high eccentricities (see for instance lines 418-423 or lines 440-445). The introduction presents this bias in the first paragraph given that most research has been conducted exploring central and near-peripheral locations. After that, we introduce the point that in naturalistic environments we orient to faces located further out in the visual field. This mixed evidence beyond near-peripheral locations is present in the adult literature and we aimed to investigate this during infancy.

  1. There are no control (non-face) experiments in the paper, so its scope is limited to peripheral visual field testing in infants. Moreover, the paper doesn't elaborate on the mechanism of non-foveal stimuli perception (lines 20-21) except for a mention of detection performance.

We have responded to the control query in the prior point above (please see response to point 2). Additionally, non-foveal stimuli perception is presented in the introduction, lines 41-82, 85-88, 96-99 and extensively in the discussion, lines 364-402.

  1. It is recommended that the paper limit its claims and scope proportionally to the results and experiment design.

We thank the Reviewer for raising this point. We have limited a claim on line 356. We have also made it clear that the requested control study has been conducted and published.

Round 2

Reviewer 3 Report

  1. All of my comments have been addressed by the authors.
  2. In order to improve the readability and make the science more accessible to the reader, it is recommended authors combine both the main experiment and the controls into a single manuscript.

Author Response

We thank the Reviewer for stating that all their comments have been addressed.

Regarding the recommendation of combining our past experiment with non-social stimuli into this manuscript, it is not possible because these data have been already published in another paper.